# Allometric Expression of Sesame Cultivars in Response to Various Environments and Nutrition

**Meer Muhammad Khan** [1], **Abdul Manaf** [1,*], **Fayyaz ul Hassan** [1], **Muhammad Sheeraz Ahmad** [2], **Abdul Qayyum** [3], **Zahid Hussain Shah** [4], **Hameed Alsamadany** [5], **Seunghwan Yang** [6] **and Gyuhwa Chung** [6,*]

1   Department of Agronomy, PMAS-Arid Agriculture University Rawalpindi, Rawalpindi 46300, Pakistan; meermuhammad@yahoo.com (M.M.K.); fayyaz.sahi@uaar.edu.pk (F.u.H.)
2   Department of Biochemistry, PMAS-Arid Agriculture University Rawalpindi, Rawalpindi 46300, Pakistan; dr.sheeraz@uaar.edu.pk
3   Department of Agronomy, The University of Haripur, Haripur 22620, Pakistan; aqayyum@uoh.edu.pk
4   Department of Plant Breeding and Genetics, PMAS Arid Agriculture University, Rawalpindi 46300, Pakistan; shahzahid578@hotmail.com
5   Department of Biological Sciences, King Abdulaziz University, Jeddah 21589, Saudi Arabia; halsamadani@kau.edu.sa
6   Department of Biotechnology, Chonnam National University, Yeosu 59626, Korea; ymichigan@chonnam.ac.kr
*   Correspondence: drmunaf@uaar.edu.pk (A.M.); chung@chonnam.ac.kr (G.C.)

**Abstract:** Sesame cultivars perform differently in various environments comprising specific locations and years. Micronutrient fertilizers increase crop yields and can enhance resistance to environmental hazards and contribute to potential yield. For assessing the effect of environment and micronutrients, field experiments were carried out at three different locations (BARI, URF and NARC) of Pothwar region, Pakistan, in two succeeding years (2014, 2015). Four sesame cultivars and three micronutrients (Zn, B and Mn) with a control treatment were used in RCB design with a split-plot procedure and four replications. The results showed significant differences in the main effects of all tested factors (cultivars, micronutrients, locations and years). Maximum mean values of plant height, capsules plant$^{-1}$, biomass yield and seed yield were recorded for SG-30, while TH-6 was the lowest for these parameters. Maximum mean values of capsule length, capsule weight, seeds capsules$^{-1}$ and the weight of 1000 seeds were recorded for TH-6 but vice versa for SG-30. Maximum capsule length and seeds capsule$^{-1}$ were observed at NARC, while other growth and yield attributes were maximum at BARI and URF, including biomass yield and seed yield. The interactive effect of cultivar × location × year was highly significant for all growth and yield traits except capsule length. Sesame cultivars revealed a differential response for all traits at three locations during both years. Application of micronutrients significantly augmented all growth and yield features of sesame. Micronutrient fertilizers (i.e., ZnSO$_4$ at 10 kg ha$^{-1}$, borax at 10 kg ha$^{-1}$ and MnSO$_4$ at 5 kg ha$^{-1}$) would increase yield and yield-attributing traits in low- or medium-precipitation areas where suitable cultivars have been designated.

**Keywords:** locations; micronutrients; rainfall; seed yield

## 1. Introduction

Pakistan is greatly deficient in edible oil production. During 2020–2021 (July–March), the country's demand for edible oil was 3.291 million tons, whereas 0.374 million tons (11.4%) were locally produced, while the remaining 2.917 million tons (88.6%) of edible oil was covered by imports [1]. Therefore, an increase in the local production of oilseeds is needed. The Pothwar plateau is a key part of the subtropical dry land zone, which extends across latitude 32°10 to 34°9 N and longitude 71°10 to 73°55 E, with 1.82 million hectares in Pakistan. The parent material of Pothwar plateau soils is varied in nature, and the parent material is loess, alluvium, colluviums and mixed by nature. The soils have variable parent material and chemical composition, and are prone to erosion due to undulated landforms.

The rainfall is erratic and differs greatly from 400 mm in the southwest to 1500 mm in the northeast part of the region. Pothwar is categorized as semi-arid to humid [2] and more than 70% of annual precipitation falls in the summer months [3]. The people of Pothwar have conventionally looked to rainfed agriculture for their livelihood. Traditional oilseeds such as sesame, which has been grown for centuries, need to be more focused on and explored to minimize the dependency on imports of edible oil.

Sesame (*Sesamum indicum* L.) seed comprises oil (50%), protein (up to 25%), crude fiber (3.2%), carbohydrate (16–18%) and ash (5.7%), and is rich source of vitamin E, phosphorus and calcium [4,5]. Sesame is known as the queen of oilseeds because of its high level of polyunsaturated fatty acids such as oleic acid (29.3 to 41.4%) and linoleic acid (40.7 to 49.3%) [6]. There is a huge gap between the potential production and actual yield of sesame in this country. Several reasons, such as meagre input management (little or no fertilization), macro- and micronutrient deficiencies in the soils, and biotic and abiotic stresses, cause low production of sesame.

Several research studies on sesame and other oilseed crops have specified that there is a diverse range of differences between genotypes and environments [7]. The yield of genotypes is uneven and differs over locations and environmental variability; specifically, location effects, seasonal fluxes and their interaction strongly impact the phenology of genotypes concerning optimum production [8]. Optimizing the yield and quality of the crop and more economic profits can be obtained through nutrient fertilization, which can also minimize the environmental risks. The idea of integrated plant nutrition is to maintain the soil fertility for high crop productivity by using combined and cheap sources such as fertilizers with the plant-required nutrients at the optimum level [9,10]. Farmers do not often use micronutrient fertilizers for crop production under rainfed conditions; therefore, micronutrient deficiencies may occur, and lower soil moisture reduces the nutrients' availability to plants [11]. Deficiencies of micronutrients in crops have increased owing to soil erosion, exhaustive farming, micronutrient losses through leaching and using marginal lands [12]. The main obstacle lies in the physiological problems associated with micronutrient deficiencies, such as hormonal imbalances that lead to reduced sesame yields [13]. Furthermore, boron, zinc and manganese help plants in chlorophyll formation and increase the photosynthetic activity [14]. Several studies have documented the increase in the growth and yield attributes of sesame due to micronutrients such as boron, zinc and manganese [15–17]. B, Zn, and Mn deficiencies were observed in the soils of the Pothwar plateau [18].

This study aimed to examine the response of sesame cultivars to the different environments of Pothwar Pakistan and to evaluate the effect of Zn, B and Mn on the growth and yield features of sesame.

## 2. Materials and Methods

### 2.1. Field Experimentation and Treatments

Three experiments were carried out: one each at Barani Agricultural Research Institute (BARI) Chakwal (latitude 32.9° N; longitude 72.8° E; elevation, 523 m; annual rainfall, 450–550 mm); University Research Farm (URF), Chakwal Road, Rawalpindi (latitude 33.1° N; longitude 73.0° E; elevation, 510 m; annual rainfall, 650–850 mm) and the National Agricultural Research Center, Islamabad (NARC) (latitude 33.7° N; longitude 73.1° E; elevation, 504 m; annual rainfall, 1000–1200 mm). The experiments were performed by using 4 sesame cultivars (SG-30, TS-3, TS-5 and TH-6) and 3 micronutrients, namely zinc, boron and manganese, with 1 control treatment. Experiments were established in a randomized complete block design (RCBD) with four replicates. All plots were randomized in each replicate, with cultivars allotted in the main plots and micronutrients in the subplots. Crops were planted in July 2014 and 2015 with single-row hand drill in a plot size of 2.7 m × 5 m, with six rows in each plot placed 45 cm apart. A 5 kg ha$^{-1}$ seed rate was kept, with a seeding depth of 3–4 cm. Recommended doses of fertilizers (N at 90 kg ha$^{-1}$ and P$_2$O$_5$ at 60 kg ha$^{-1}$) were applied as urea and di-ammonium phosphate

(DAP) during the last plowing at all locations. All cultivation operations remained the same at each site. $ZnSO_4$ (10 kg ha$^{-1}$), borax (10 kg ha$^{-1}$) and $MnSO_4$ (5 kg ha$^{-1}$) were applied as the Zn, B and Mn sources, respectively. The side banding method was used for micronutrient application just after sowing across all locations. Manual thinning was carried out after the complete emergence to maintain the distance between plants at 10 cm.

### 2.2. *Soil and Weather Data*

Composite soil samples at two depths (0–15 cm and 15–30 cm) from all locations were collected before sowing and the samples were analyzed for physiochemical properties (Table 1). Zn and Mn were examined by DTPA extraction [19], while B was measured colorimetrically by means of azomethine—H [20] after hot water extraction [21]. Weather data of the experimental sites in both years were obtained from Pakistan's Meteorological Department, Islamabad (Figure 1).

**Table 1.** Soil physio-chemical analysis of the experimental sites during 2014 and 2015.

| | BARI (Chakwal) | | | | URF (Rawalpindi) | | | | NARC (Islamabad) | | | |
|---|---|---|---|---|---|---|---|---|---|---|---|---|
| | 2014 | | 2015 | | 2014 | | 2015 | | 2014 | | 2015 | |
| Depth (cm) | 0–15 | 15–30 | 0–15 | 15–30 | 0–15 | 15–30 | 0–15 | 15–30 | 0–15 | 15–30 | 0–15 | 15–30 |
| Soil texture | SL | SL | SL | SL | SCL | SCL | SCL | SCL | SCL | SCL | SCL | SCL |
| pH | 7.78 | 7.77 | 7.75 | 7.73 | 7.84 | 7.81 | 7.91 | 7.82 | 7.61 | 7.59 | 7.69 | 7.65 |
| OM (%) | 0.83 | 0.80 | 0.77 | 0.75 | 0.75 | 0.72 | 0.81 | 0.79 | 1.07 | 1.02 | 1.1 | 0.98 |
| EC (dS m$^{-1}$) | 0.39 | 0.38 | 0.51 | 0.49 | 0.61 | 0.58 | 0.59 | 0.56 | 0.81 | 0.79 | 0.8 | 0.79 |
| BD (g cm$^{-3}$) | 1.47 | 1.49 | 1.46 | 1.47 | 1.53 | 1.55 | 1.49 | 1.5 | 1.39 | 1.4 | 1.38 | 1.39 |
| Porosity (%) | 44.5 | 43.8 | 44.9 | 44.5 | 42.3 | 41.5 | 43.8 | 43.4 | 47.5 | 47.2 | 47.9 | 47.5 |
| NO$_3$ N (mg kg$^{-1}$) | 5.61 | 5.37 | 5.5 | 5.43 | 3.74 | 3.68 | 4.01 | 3.95 | 7.3 | 7.29 | 7.36 | 7.32 |
| P (mg kg$^{-1}$) | 4.52 | 4.33 | 4.7 | 4.65 | 3.79 | 3.74 | 3.88 | 3.82 | 4.13 | 4.09 | 4.19 | 3.98 |
| K (mg kg$^{-1}$) | 139 | 130 | 143 | 137 | 126 | 120 | 136 | 128 | 118 | 117 | 130 | 124 |
| Zn (mg kg$^{-1}$) | 0.41 | 0.38 | 0.43 | 0.39 | 0.46 | 0.45 | 0.42 | 0.41 | 0.26 | 0.25 | 0.32 | 0.29 |
| Mn (mg kg$^{-1}$) | 1.15 | 1.09 | 1.03 | 0.98 | 1.2 | 1.16 | 1.19 | 1.12 | 1.16 | 1.11 | 1.02 | 0.97 |
| B (mg kg$^{-1}$) | 0.39 | 0.36 | 0.4 | 0.37 | 0.43 | 0.42 | 0.44 | 0.41 | 0.36 | 0.33 | 0.35 | 0.31 |

SL = sandy loam, SCL = sandy clay loam, OM = organic matter, EC = electric conductivity, BD = bulk density.

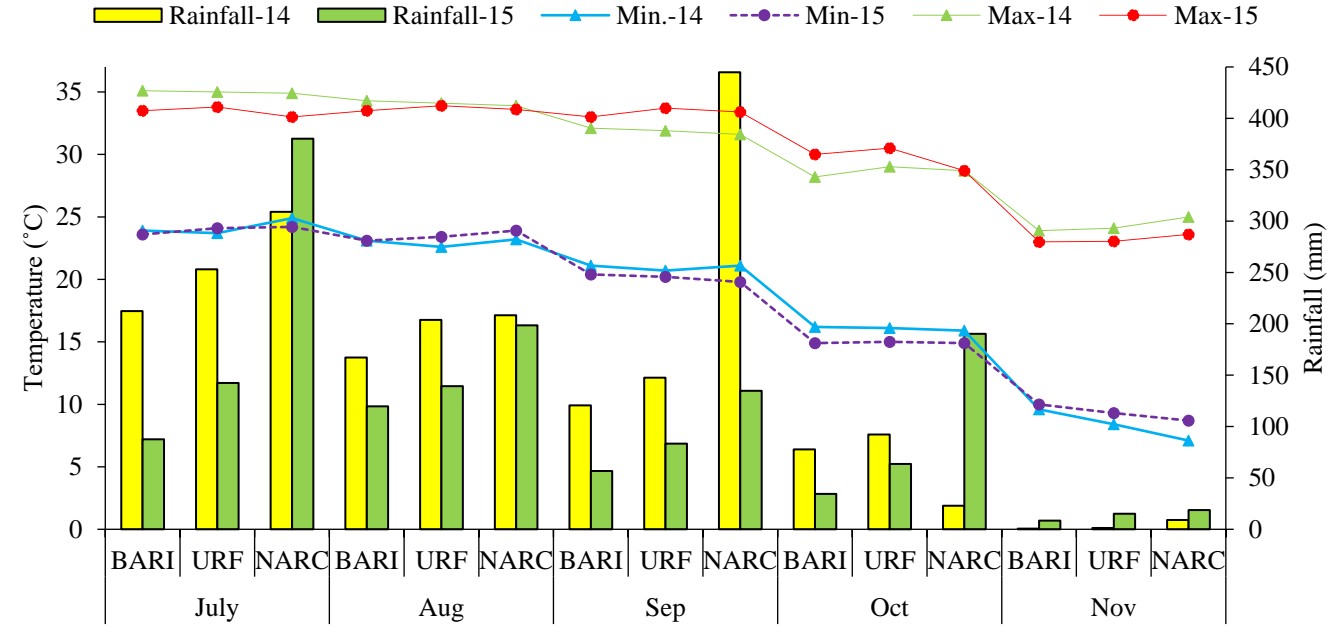

**Figure 1.** Agro-meteorological data of the experimental sites during 2014 and 2015.

### 2.3. Crop Data

Crop data were collected on the growth and yield-attributing characteristics of sesame. Data for plant height (PH) (using a measuring tape), number of primary branches (NBP), secondary branches (NSB) and number of capsules (NC) were obtained from 10 randomly selected plants in each plot at the time of maturity, and the per-plant average was calculated. Data for capsule length (CL) and capsule weight (CW) were recorded from 20 capsules, which were randomly collected from plants in each plot. Firstly, the length of each capsule was measured by using a ruler scale, and the average capsule length was worked out. Secondly, the weight of a sample of 20 capsules was measured by using an electric balance, and the mean capsule weight was calculated. One hundred capsules of mature plants in each plot were collected randomly. After manual threshing of the capsules obtained, the total number of seeds was counted with the help of a seed counter, and the mean number of seeds per capsule (SPC) was calculated. Mature plants in the 2 central rows of a 2 m row length were manually harvested from each plot, and the tagged bundles of plants were kept vertical in the field. After few days of sun drying, the biomass yield (BioY) of each plant bundle was recorded using a weight balance, and the results were converted into kg ha$^{-1}$. Later, the plant bundles were threshed manually, and clean seeds were obtained by removing plant debris. The weight of each seed sample was recorded using an electric balance, and the seed yield (SY) in kg ha$^{-1}$ was calculated. Three batches of 1000 seeds were isolated from the bulk seed sample obtained from each plot by using a seed counter, the weight of each batch was recorded utilizing an electric balance, and the average weight of 1000 seeds (SW1000) was calculated.

### 2.4. Statistical Analysis

A general linear model was adopted for analysis of variance (ANOVA) using a computer program (statistix 8.1). An RCB design (factorial) was used to determine the significance level of the main experimental factors (cultivars, micronutrients, locations and years) and their possible interactions. Furthermore, to identify significant variation between the means, the least significance test (LSD) was applied with a probability level of 5%. The linear association between the agronomic traits were determined by Pearson's correlation, and the correlation coefficient (*r*) values [22] along with the *p*-values were computed using statistix 8.1. Additionally, principal component analysis (PCA) [23] was executed using R software (Version 4.0.0) to classify numerous agronomic traits of sesame in response to the treatment interactions (cultivar × location × year).

## 3. Results

### 3.1. Cultivars

The main effects of sesame cultivars for observed growth and yield characteristics were found to be highly significant at *p* < 0.01 (Table 2). Maximum values of PH, NC, biomass yield and seed yield (725.0 kg ha$^{-1}$) were observed for the cultivar SG-30, but this was lowest for CL, CW, SPC and 1000 SW. Maximum NPB and NSB were observed for TS-3. On the other hand, TH-6 had the highest values for CL, CW, SPC and 1000 SW. However, it was lowest for the remaining attributes including seed yield (363.4 kg ha$^{-1}$). The cultivar TS-5 was statistically on par with SG-30 for seed yield, while TS-3 was statistically parallel with SG-30 for seeds capsule$^{-1}$, and capsule length and weight (Table 2).

**Table 2.** Main and interactive effects of cultivars, micronutrients, locations and years for growth and yield attributes of sesame.

| | Plant Height (cm) | No. of Primary Branches | No. of Secondary Branches | Capsules Plant$^{-1}$ | Capsule Length (cm) | Capsule Weight (g) | Seeds Capsule$^{-1}$ | 1000 Seed Weight (g) | Biomass Yield (kg ha$^{-1}$) | Seed Yield (kg ha$^{-1}$) |
|---|---|---|---|---|---|---|---|---|---|---|
| Cultivar (C) | ** | ** | ** | ** | ** | ** | ** | ** | ** | ** |
| SG-30 | 118.4 a | 5.50 c | 3.74 c | 55.6 a | 2.56 c | 0.263 c | 54.7 b | 2.35 d | 3281 a | 725.0 a |
| TS-3 | 114.4 b | 6.72 a | 5.15 a | 47.2 c | 2.58 c | 0.265 c | 54.2 b | 2.46 c | 3141 b | 647.2 b |
| TS-5 | 114.1 b | 5.87 b | 4.54 b | 54.2 b | 2.64 b | 0.273 b | 57.8 a | 2.54 b | 2894 c | 734.6 a |
| TH-6 | 92.3 c | 0.47 d | 0.09 d | 26.5 d | 3.41 a | 0.341 a | 58.4 a | 2.86 a | 1144 d | 363.4 c |
| SE± | 1.11 | 0.05 | 0.051 | 0.65 | 0.025 | 0.004 | 0.378 | 0.022 | 54.65 | 13.01 |
| Locations (L) | ** | ** | ** | ** | ** | ** | ** | ** | ** | ** |
| BARI | 115.1 b | 4.07 b | 2.55 c | 47.0 b | 2.82 b | 0.291 a | 55.0 b | 2.70 a | 2614 b | 660.0 a |
| URF | 123.0 a | 5.74 a | 4.58 a | 47.4 b | 2.74 b | 0.295 a | 55.1 b | 2.68 a | 3161 a | 632.5 b |
| NARC | 91.3 c | 4.10 b | 3.01 b | 43.1 a | 2.83 a | 0.272 b | 58.7 a | 2.28 b | 2071 c | 560.1 c |
| SE± | 0.96 | 0.043 | 0.044 | 0.57 | 0.021 | 0.0034 | 0.327 | 0.019 | 47.33 | 11.27 |
| Year (Y) | ** | ** | ** | ** | NS | ** | NS | ** | * | NS |
| 2014 | 111.4 a | 4.76 a | 3.77 a | 43.6 | 2.81 | 0.274 b | 56.5 | 2.49 b | 2657 a | 617.0 |
| 2015 | 108.2 b | 4.55 b | 2.99 b | 48.1 | 2.78 | 0.297 a | 56.1 | 2.61 a | 2573 b | 618.1 |
| SE± | 0.78 | 0.035 | 0.036 | 0.48 | 0.018 | 0.0028 | 0.267 | 0.015 | 38.65 | 9.20 |
| Micronutrients (MN) | ** | ** | ** | ** | ** | ** | ** | ** | ** | ** |
| Control | 105.7 c | 4.33 d | 3.08 c | 41.9 d | 2.71 c | 0.271 c | 55.2 c | 2.48 c | 2408 d | 563.1 c |
| Zinc | 111.9 a | 4.89 a | 3.61 a | 47.0 b | 2.84 a | 0.293 a | 56.8 ab | 2.58 a | 2858 a | 642.0 a |
| Boron | 112.5 a | 4.78 b | 3.56 a | 49.2 a | 2.86 a | 0.294 a | 57.0 a | 2.60 a | 2657 b | 657.7 a |
| Manganese | 109.1 b | 4.56 c | 3.27 b | 45.4 c | 2.79 b | 0.284 b | 56.1 b | 2.54 b | 2537 c | 607.4 b |
| SE± | 1.11 | 0.05 | 0.051 | 0.65 | 0.025 | 0.004 | 0.378 | 0.022 | 54.65 | 13.01 |
| Y × L | ** | ** | ** | ** | ** | ** | * | ** | ** | ** |
| Y × C | ** | ** | ** | ** | NS | ** | NS | ** | * | ** |
| Y × MN | NS | NS | NS | NS | NS | NS | NS | NS | NS | NS |
| L × C | ** | ** | ** | ** | ** | NS | ** | ** | ** | ** |
| L × MN | NS | NS | ** | NS | NS | NS | NS | NS | NS | NS |
| C × MN | NS | ** | ** | NS | * | NS | NS | NS | NS | NS |
| Y × L × C | ** | ** | ** | ** | NS | ** | ** | ** | ** | ** |
| Y × L × MN | NS | NS | NS | NS | NS | NS | NS | NS | NS | NS |
| Y × C × MN | NS | NS | NS | NS | * | NS | NS | NS | NS | NS |
| L × C × MN | NS | NS | NS | NS | NS | NS | NS | NS | NS | NS |
| Y × L × C × MN | NS | NS | NS | NS | NS | NS | NS | NS | NS | NS |

** Significant at $p \leq 0.01$. * Significant at $p \leq 0.05$. NS = Non-significant. Values in columns having the different letter are significantly different at $p \leq 0.05$.

### 3.2. Locations

Location means revealed statistically significant differences ($p < 0.01$) for all studied traits (Table 2). BARI achieved the maximum 1000 SW and seed yield (660 kg ha$^{-1}$), while URF produced sesame plants with the maximum PH, NPB, NSB, capsules plant$^{-1}$ (47.4), CW and biomass yield (3161 kg ha$^{-1}$). Although URF and BARI were statistically on par for the locational means of capsules plant$^{-1}$, CW, SPC and 1000 SW, NARC had the maximum results for CL and seeds capsule$^{-1}$, but this location produced sesame plants with minimum height, NPB, NSB, CW, 1000 SW, biomass yield (2071 kg ha$^{-1}$) and seed yield (597.6 kg ha$^{-1}$).

### 3.3. Year

Year means showed differential responses for the observed parameters, except CL, SPC and seed yield (Table 2), where both years revealed statistically parallel ($p > 0.05$) results for these traits. However, PH, NPB, NSB and biomass yield (2657 kg ha$^{-1}$) were significantly higher during the first year (2014). However, capsules plant$^{-1}$, CW and 1000 SW were significantly ($p < 0.01$) superior during second year (2015).

### 3.4. Cultivar × Location × Year Interactions

The three-way interactions of year, location and cultivar for the yield traits of sesame were highly significant ($p < 0.01$) for the recorded growth and yield parameters (except capsule length) (Table 2). Graphical presentations of the interactions (showing significant differences) for quantitative characteristics of sesame are shown in Figures 2–10.

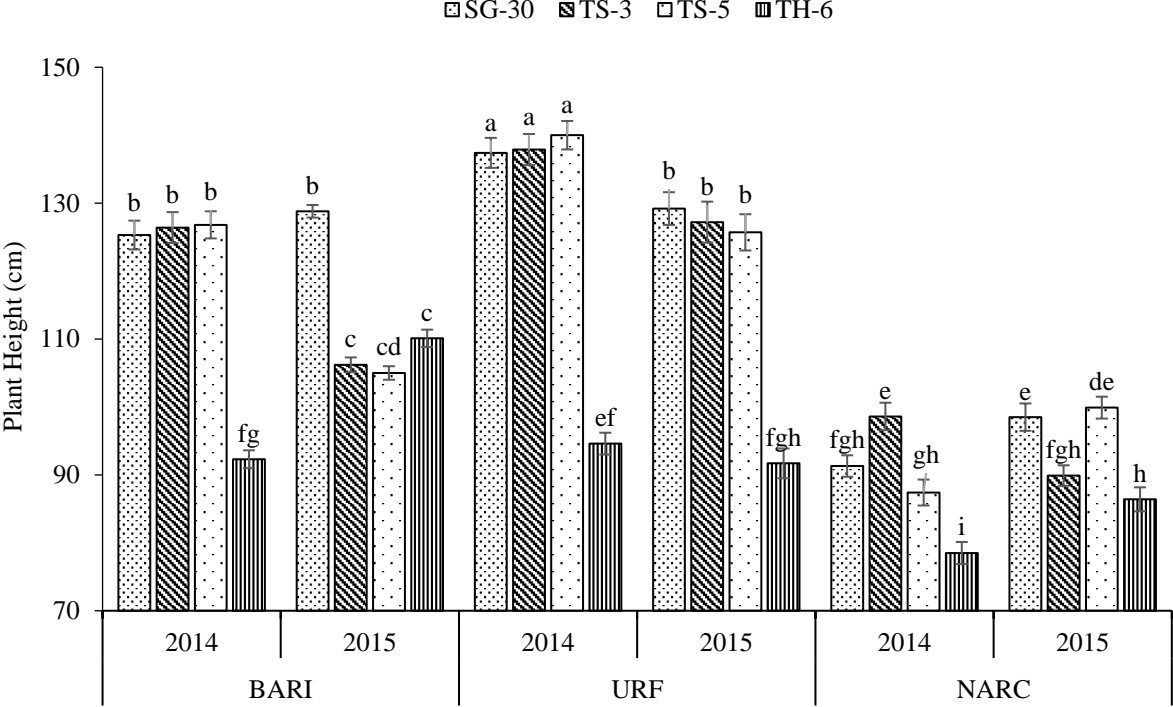

**Figure 2.** Plant height (cm) of four sesame cultivars at three locations during 2014 and 2015 (pooled over micronutrients). The bars indicates standard error (±SE) of mean (n = 3). All means are significantly different at $p \leq 0.05$.

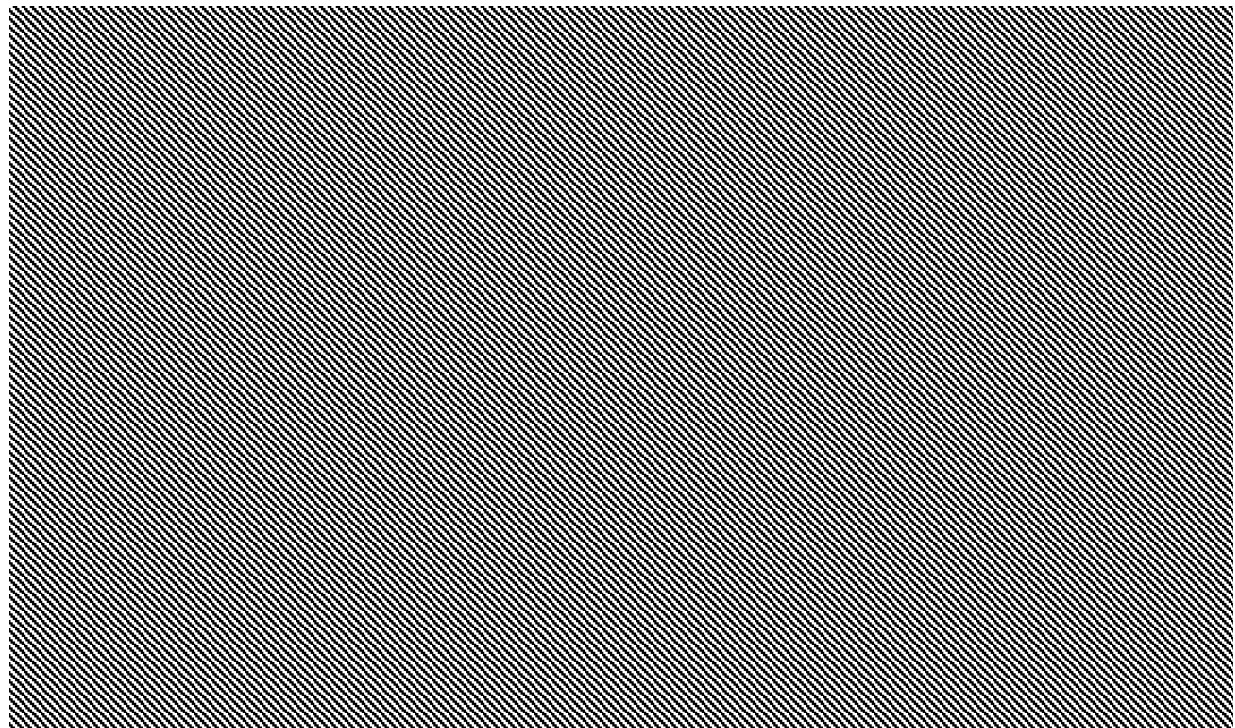

**Figure 3.** Number of primary branches plant$^{-1}$ of four sesame cultivars at three locations during 2014 and 2015 (pooled over micronutrients). The bars indicates standard error (±SE) of mean (n = 3). All means are significantly different at $p \leq 0.05$.

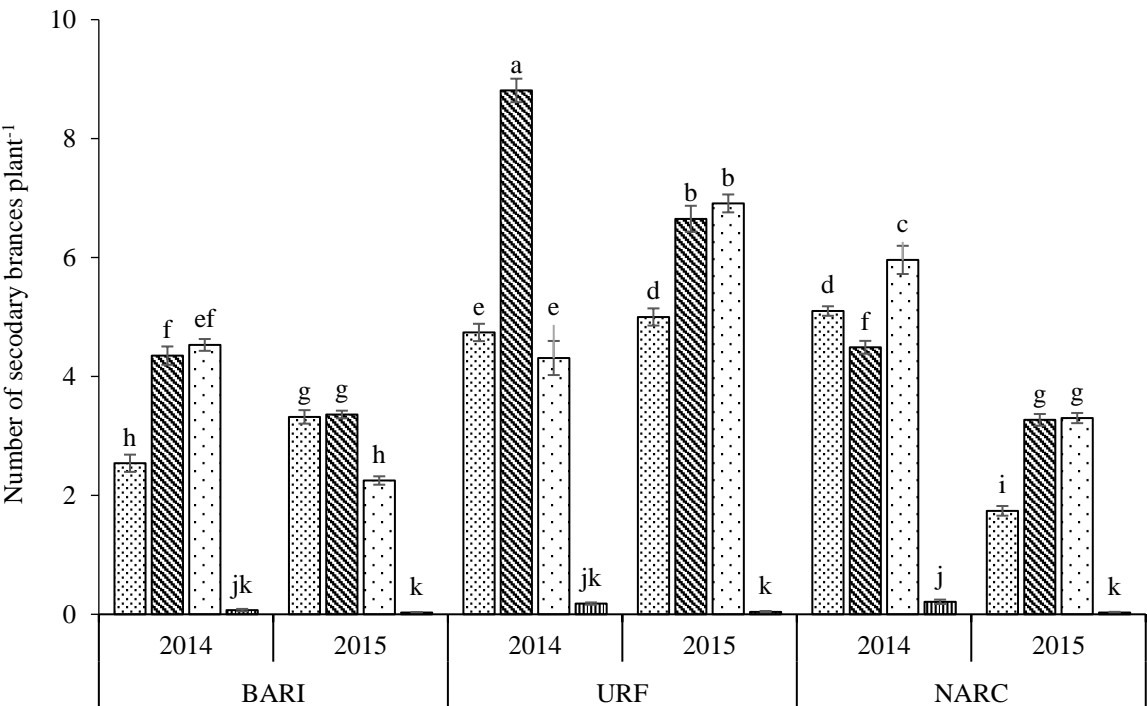

**Figure 4.** Number of secondary branches plant$^{-1}$ of four sesame cultivars at three locations during 2014 and 2015 (pooled over micronutrients). The bars indicates standard error (±SE) of mean (n = 3). All means are significantly different at $p \leq 0.05$.

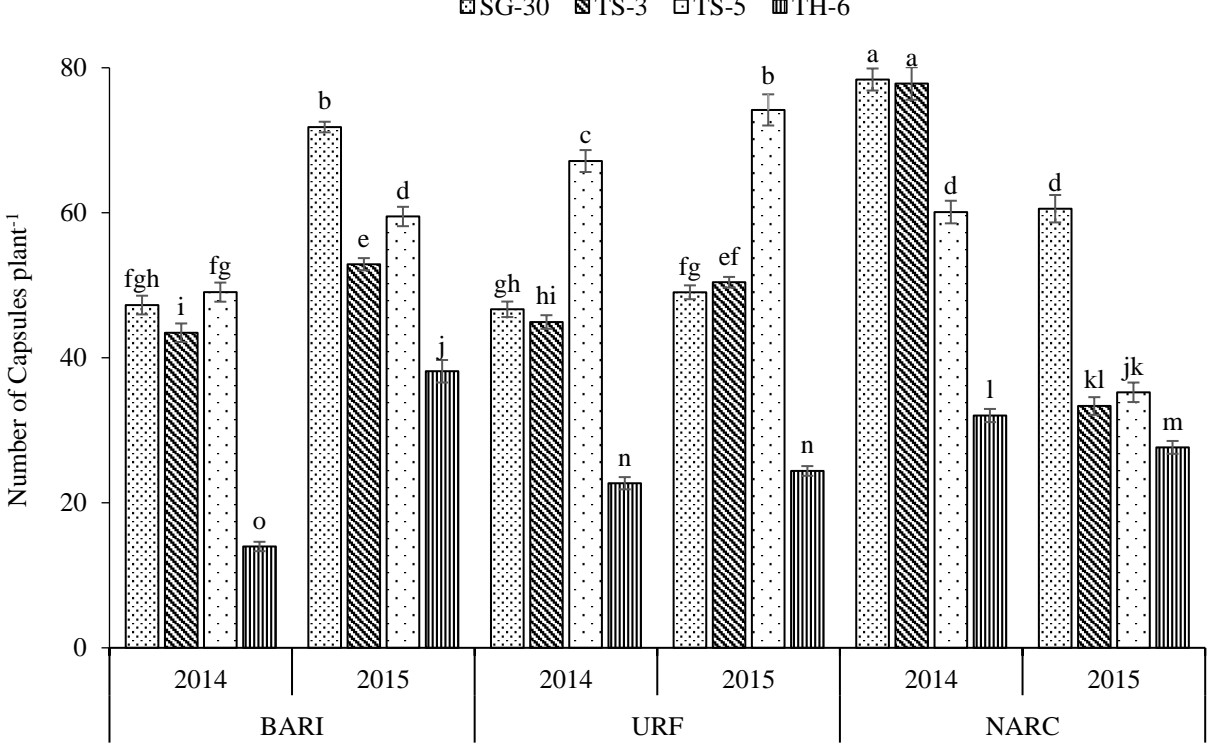

**Figure 5.** Capsules plant$^{-1}$ of four sesame cultivars at three locations during 2014 and 2015 (pooled over micronutrients). The bars indicates standard error (±SE) of mean (n = 3). All means are significantly different at $p \leq 0.05$.

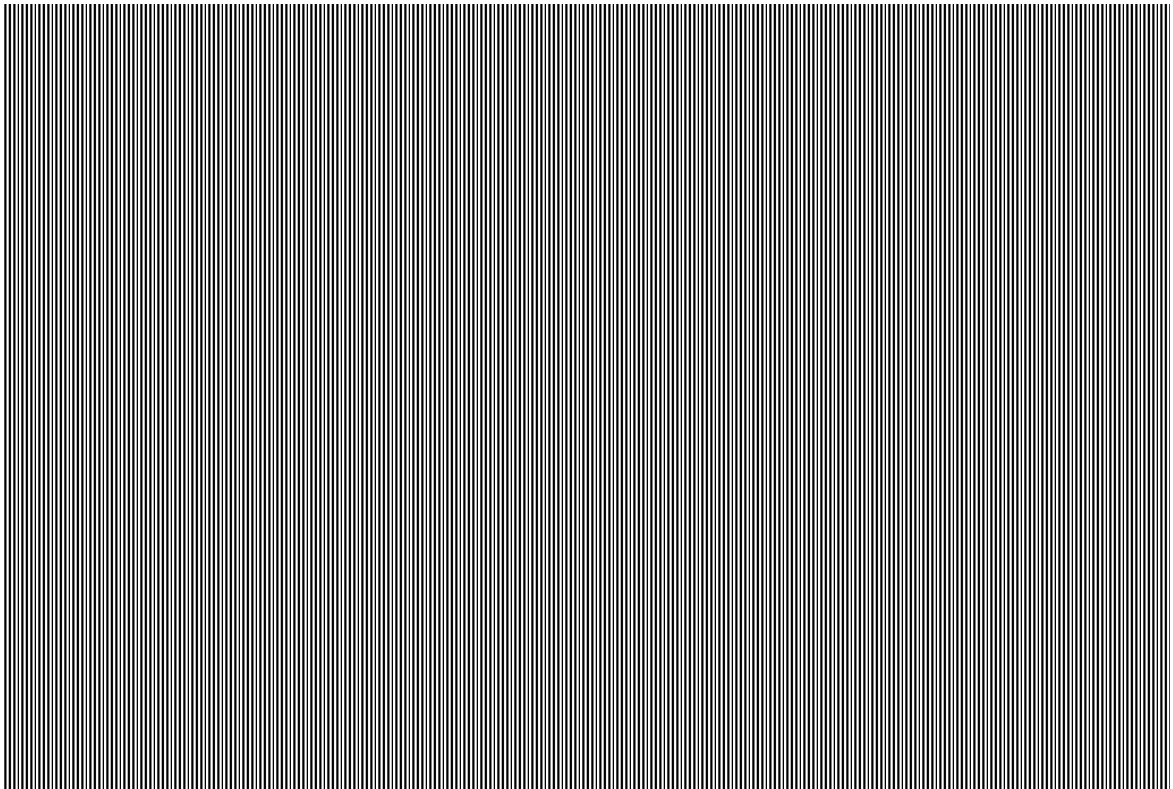

**Figure 6.** Capsule weight (g) of four sesame cultivars at three locations during 2014 and 2015 (pooled over micronutrients). The bars indicates standard error (±SE) of mean (n = 3). All means are significantly different at $p \leq 0.05$.

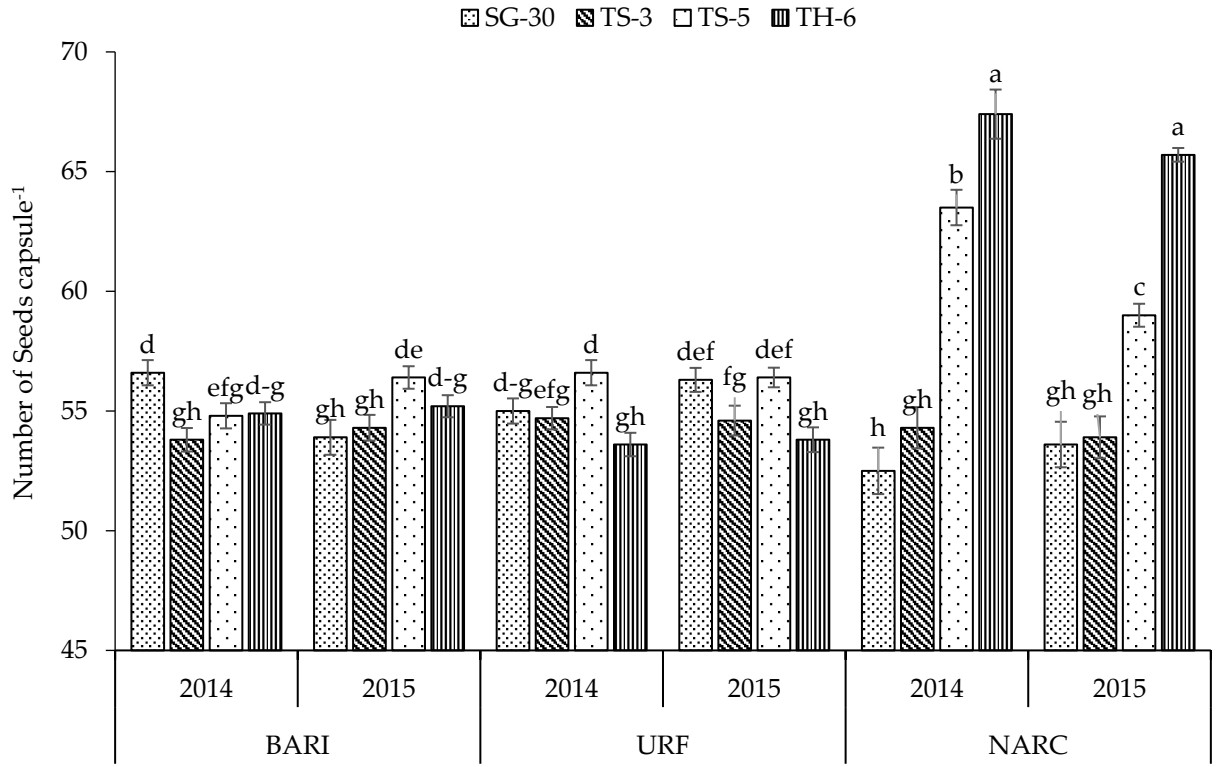

**Figure 7.** Number of seeds capsule$^{-1}$ of four sesame cultivars at three locations during 2014 and 2015 (pooled over micronutrients). The bars indicates standard error (±SE) of mean (n = 3). All means are significantly different at $p \leq 0.05$.

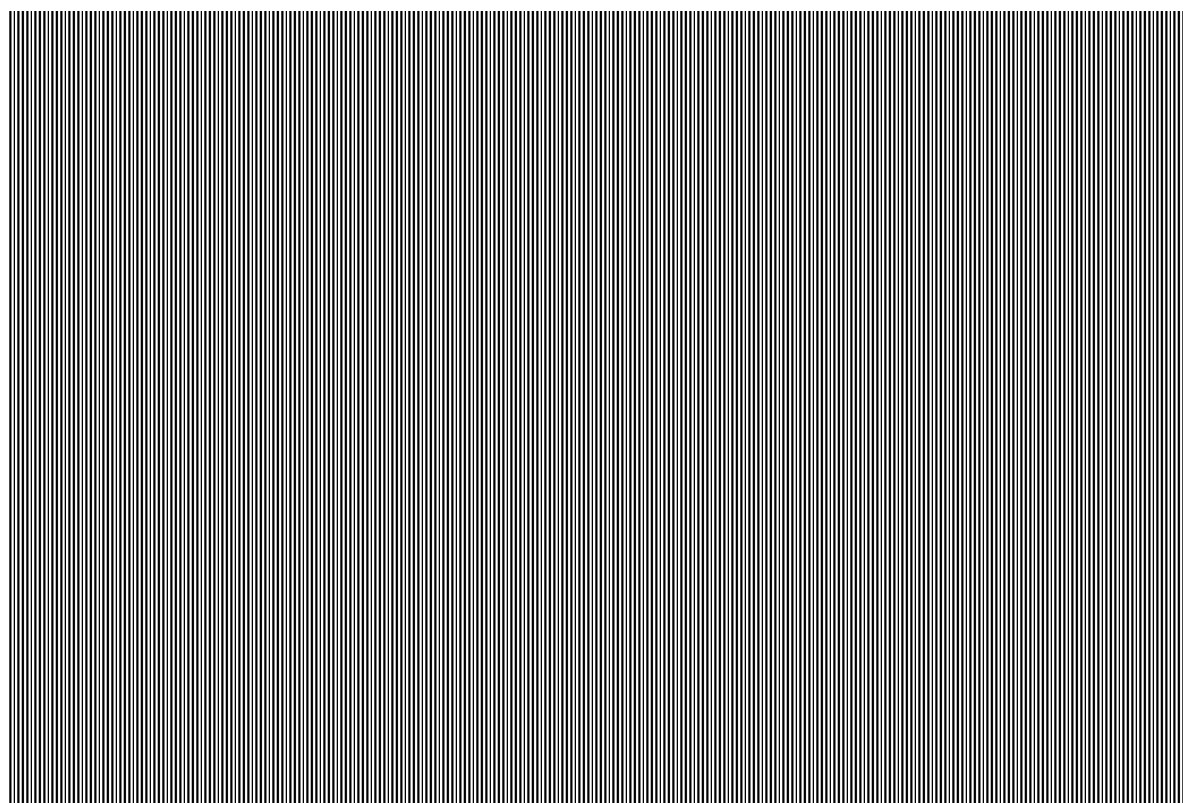

**Figure 8.** Thousand seed weight (g) of four sesame cultivars at three locations during 2014 and 2015 (pooled over micronutrients). The bars indicates standard error (±SE) of mean (n = 3). All means are significantly different at $p \leq 0.05$.

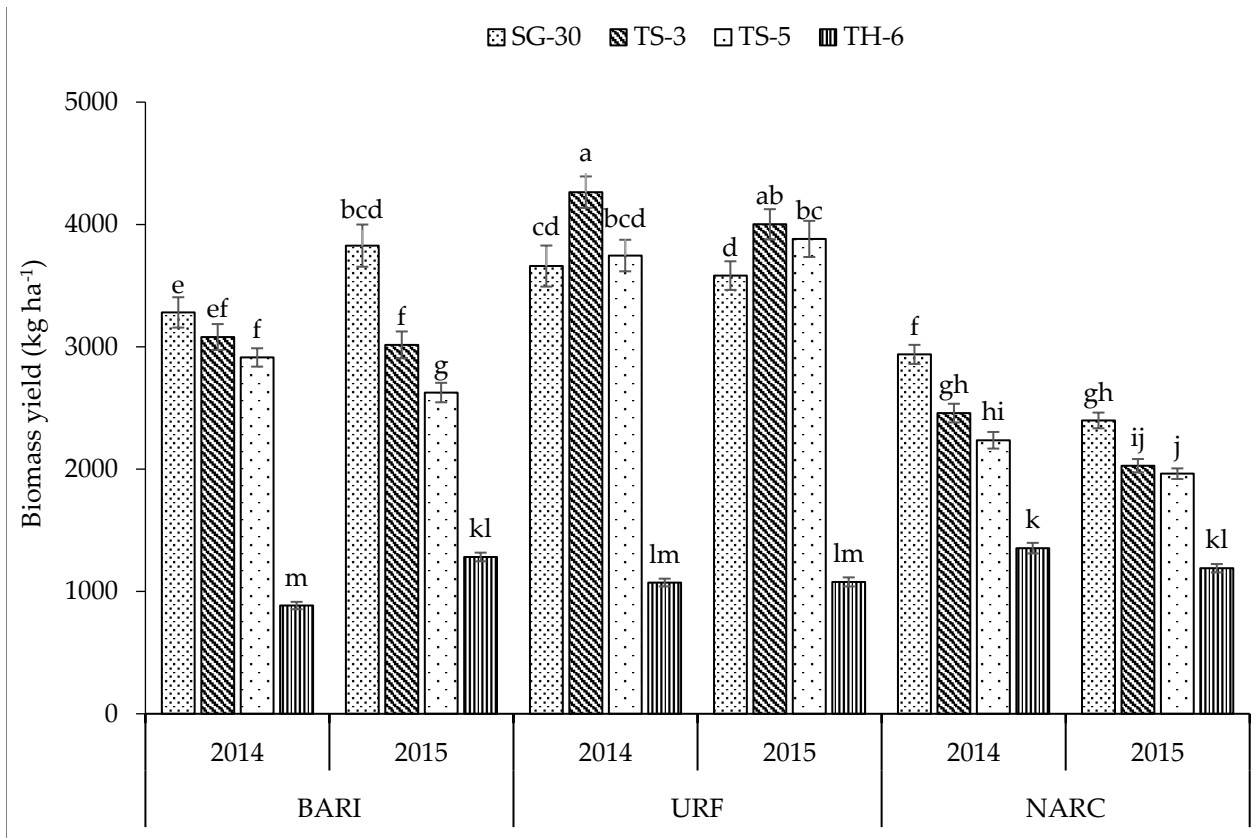

**Figure 9.** Biomass yield (kg ha$^{-1}$) of four sesame cultivars at three locations during 2014 and 2015 (pooled over micronutrients. The bars indicates standard error (±SE) of mean (n = 3). All means are significantly different at $p \leq 0.05$.

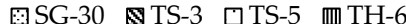

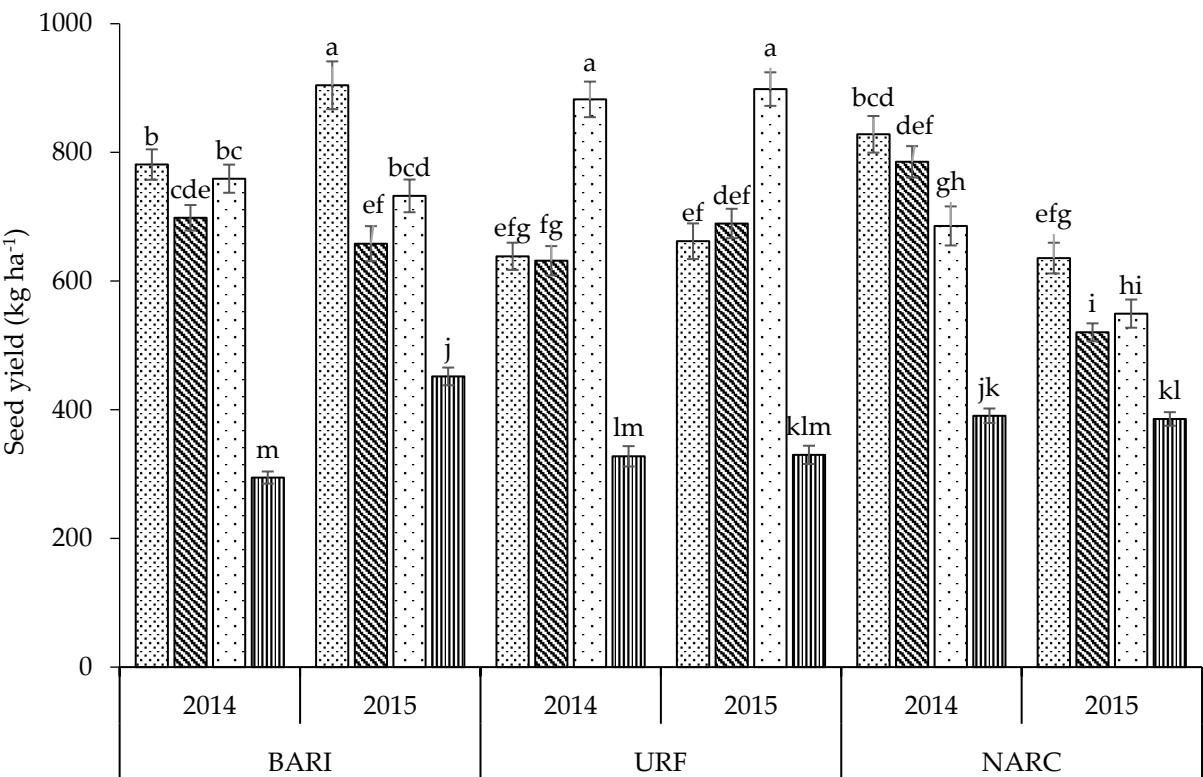

**Figure 10.** Seed yield (kg ha$^{-1}$) of four sesame cultivars at three locations during 2014 and 2015 (pooled over micronutrients). The bars indicates standard error ($\pm$SE) of mean (n = 3). All means are significantly different at $p \leq 0.05$.

The data represented in Figure 2 indicated that plant height was maximum for TS-5 (140 cm), which was statistically parallel with TS-3 and SG-30 at URF in 2014, while minimum plant height (78.5 cm) was noted for TH-6 at NARC in 2014.

It is evident from the data that SG-30 produced the maximum (10.02) number of primary branches at BARI during 2014 (Figure 3), while TH-6 produced the minimum number of primary branches (0.14) at NARC during 2015. The maximum number of secondary branches per plant was recorded for TS-3 (8.81) at URF in 2014. However, TH-6 had similarly low results for secondary branches (0.03) at BARI and NARC during 2015 (Figure 4). The results revealed that SG-30 produced the maximum primary branches across all locations during both years. Being a single-stem cultivar, TH-6 showed the minimum results for primary and secondary branches per plant across all locations during both years. However, the location URF in 2015 was found to be more conducive for producing the maximum secondary branches.

The maximum number of capsules plant$^{-1}$ was recorded for TS-5 (74.2) at URF during 2015, which was statistically parallel with SG-30 (71.8) at BARI during the second year (Figure 5), whereas TH-6 produced the fewest capsules plant$^{-1}$ (14) at BARI during 2014. In addition, TH-6 produced the minimum capsules plant$^{-1}$ across all locations during both years, while the rest of the cultivars revealed differential responses at three locations and two years.

The data for capsule weight (g) revealed that the maximum capsule weight was observed for TH-6 at BARI (0.427 g), which was statistically on a par with URF during the second year (Figure 6). However, the minimum capsule weight (0.209 g) was noted for TS-3 at NARC during the second year. It is apparent from Figure 6 that the capsule weight of TH-6 was considerably higher across all locations during both years.

Similarly, the cultivar TH-6 achieved statistically parallel high numbers of seeds capsule$^{-1}$ at NARC, i.e., 67.4 and 65.7 in 2014 and 2015, respectively. Likewise, seeds capsule$^{-1}$ was higher for TS-5 at NARC during both years, but the lowest number of seeds capsule$^{-1}$ (52.5) was recorded for SG-30 at NARC during the first year. Other interaction means were not significantly different (Figure 7).

In the same way, TH-6 produced the maximum 1000 SW at BARI (3.67 g), followed by URF during the second year. The minimum 1000 SW was observed for SG-30 and TS-5 (statistically similar) at NARC during 2015 (1.97 g and 2.00 g, respectively). The data revealed that BARI and URF were more favorable for the seed indices of sesame (Figure 8).

Three-way interaction means for biomass yield revealed that TS-3 produced the highest biomass yield (4264 kg ha$^{-1}$) at URF in 2014, which was statistically on par with 2015. The lowest value (885 kg ha$^{-1}$) of biomass yield was reckoned for TH-6 at BARI during the first year. However, TH-6 produced the minimum biomass yield across all three locations during both years, while the rest of the cultivars revealed differential responses for biomass production (Figure 9).

The results in Figure 10 indicated that the maximum seed yield (904.3 kg ha$^{-1}$) was recorded for SG-30 at BARI in 2015, which was statistically on par with TS-5 at URF in the first and second year. However, the lowest seed yield (294.6 kg ha$^{-1}$) was observed for TH-6 at BARI during 2014. Similar to biomass yield, TH-6 produced the lowest seed yield across all locations during both years.

### 3.5. Correlations among Agronomic Traits

Seed yield revealed a significant ($p < 0.01$) and positive correlation with PH, NPB, NSB, NC and biomass yield (Table 3). Similarly, biomass yield was significantly and positively correlated with these traits. However, biomass and seed yield were negatively correlated with CL, CW, SPC and seed indices. Usually, yield-attributing traits positively contribute towards final seed yield, but in the present case, the cultivar TH-6 revealed some unique characteristics, such as longer and heavier capsules, and higher SPC and seed indices, although, because of the single stem, the shortest plants and the lowest number of capsules, it had the lowest results for biomass and seed yield. Therefore, other growth and yield-attributing traits were negatively correlated with CL, CW, SPC and seed indices.

**Table 3.** Correlation coefficients among the agronomic traits of sesame.

| Traits | PH | NPB | NSB | NC | CL | CW | SPC | 1000-SW | BioY |
|--------|-----|------|------|------|------|------|------|---------|------|
| NPB | 0.603 ** | | | | | | | | |
| NSB | 0.561 ** | 0.927 ** | | | | | | | |
| NC | 0.326 ** | 0.652 ** | 0.578 ** | | | | | | |
| CL | −0.469 ** | −0.736 ** | −0.624 ** | −0.540 ** | | | | | |
| CW | −0.195 ** | −0.379 ** | −0.311 ** | −0.077 NS | 0.556 ** | | | | |
| SPC | −0.300 ** | −0.233 ** | −0.148 ** | −0.153 ** | 0.327 ** | 0.111 * | | | |
| 1000 SW | 0.079 NS | −0.221 ** | −0.161 ** | −0.066 NS | 0.422 ** | 0.708 ** | −0.128 * | | |
| BioY | 0.755 ** | 0.869 ** | 0.796 ** | 0.620 ** | −0.667 ** | −0.288 ** | −0.218 ** | −0.142 ** | |
| SY | 0.597 ** | 0.717 ** | 0.610 ** | 0.770 ** | −0.624 ** | −0.229 ** | −0.172 ** | −0.097 NS | 0.848 ** |

** Significant at $p \leq 0.01$. * Significant at $p \leq 0.05$). NS = Non-significant.

### 3.6. Principal Component Analysis

The total eigenvalues represent the quantity of characteristics under evaluation. The first two factors denote the information contained in 5.85% and 1.78% of the original quantity of attributed traits. From the interaction of cultivars, location and years, the first three important components were basic, controlling 85.9% of the total variability (Table 4). The initial two components had the greatest influence with an aggregate contribution to the total variation of 76.3%, with an eigenvalue greater than 1. Agronomic traits including BioY, NPB and SY had the highest positive estimation in PC1, followed by 1000-SW, PH and NC in the second component's segment, whereas CL and CW had

the highest negative stacking in PC1, followed by SPC in the second principal component. Among all PCs, PC1, PC2 and PC3 contributed 58.4%, 17.8% and 9.6%, respectively, to the total diversity (Table 4).

The associations among the various traits and sesame cultivars are additionally represented by the biplot in Figure 11. The biplot illustrates that the mean performance of the treatment interaction explained 76.3% of the total variation. The vast majority of the treatment interactions (cultivar × location × year) were dispersed in all coordinates of the principal components. The sesame parameter BioY was positively associated with SY, NBP and NSB, and negatively connected with CL, CW and SPC; most of these characteristics had higher magnitudes, represented by the length of vector, and made a correspondingly greater contribution to the overall variation. A few variables displayed smaller magnitudes, with small or medium vector lengths, such as SPC, PH and NC. The traits with close vector angles presented positive correlations with each other, as exhibited by BioY, SY, NBP and NSB.

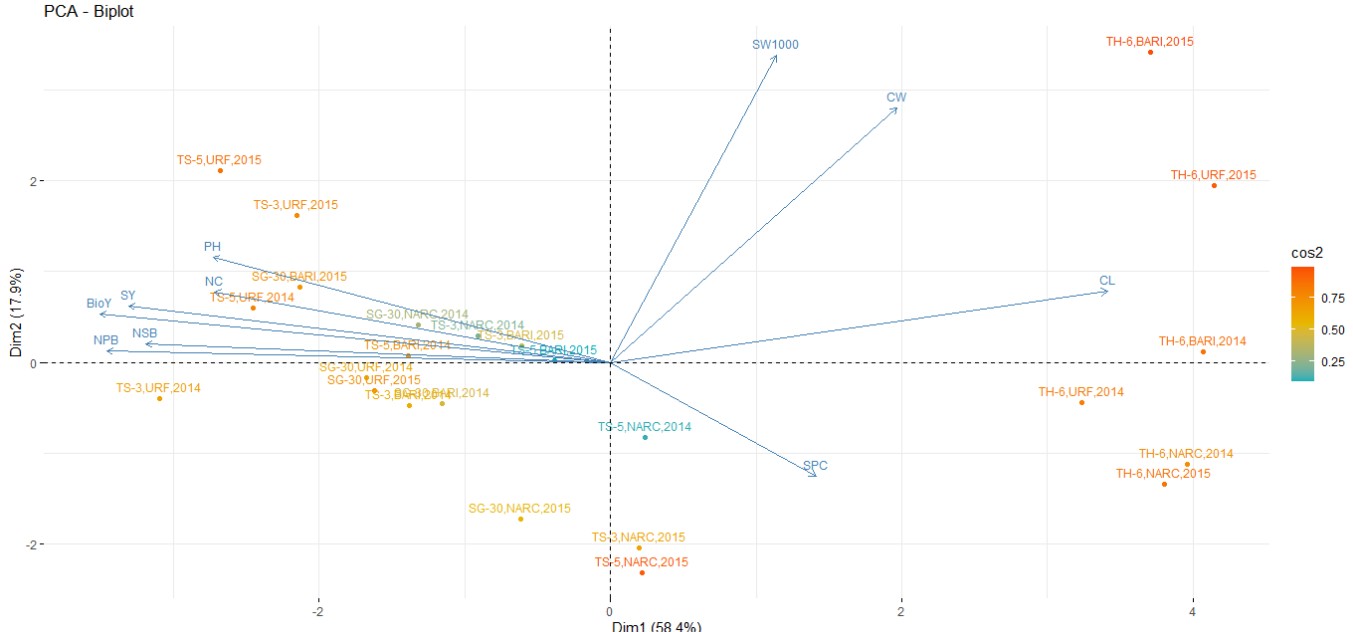

**Figure 11.** PCA biplot of the agronomic traits of sesame and treatment interactions (cultivar × location × year).

**Table 4.** Principal component matrix of the agronomic traits of sesame and treatment interactions (cultivar × location × year).

| Variable | PC1 | PC2 | PC3 | PC4 | PC5 | PC6 | PC7 | PC8 | PC9 | PC10 |
|---|---|---|---|---|---|---|---|---|---|---|
| Plant high (PH) | 0.307 | 0.237 | −0.251 | 0.518 | −0.44 | 0.232 | 0.218 | −0.328 | 0.238 | 0.242 |
| Primary branches (NPB) | 0.389 | 0.025 | 0.118 | 0.118 | 0.403 | −0.043 | −0.166 | 0.462 | 0.453 | 0.458 |
| Secondary branches (NSB) | 0.359 | 0.042 | 0.164 | 0.253 | 0.597 | −0.125 | 0.264 | −0.404 | −0.034 | −0.415 |
| Number of capsules (NC) | 0.306 | 0.158 | 0.398 | −0.583 | −0.099 | 0.103 | 0.414 | −0.186 | −0.153 | 0.365 |
| Capsule length (CL) | −0.385 | 0.16 | 0.128 | 0.183 | 0.024 | 0.054 | 0.725 | 0.454 | 0.157 | −0.129 |
| Capsule weight (CW) | −0.222 | 0.57 | 0.271 | −0.073 | 0.134 | 0.574 | −0.335 | −0.101 | 0.208 | −0.184 |
| Seeds capsule$^{-1}$ (SPC) | −0.157 | −0.253 | 0.785 | 0.419 | −0.224 | −0.122 | −0.155 | −0.105 | −0.055 | 0.128 |
| 1000 Seed weight (SW1000) | −0.129 | 0.688 | −0.023 | 0.108 | 0.02 | −0.62 | −0.103 | −0.009 | −0.252 | 0.195 |
| Biomass yield (BioY) | 0.394 | 0.11 | 0.044 | 0.223 | −0.081 | 0.306 | −0.069 | 0.461 | −0.666 | −0.142 |
| Seed yield (SY) | 0.372 | 0.128 | 0.17 | −0.193 | −0.449 | −0.303 | −0.079 | 0.203 | 0.369 | −0.549 |
| Eigenvalue | 5.84 | 1.78 | 0.96 | 0.69 | 0.43 | 0.12 | 0.07 | 0.04 | 0.02 | 0.01 |
| Proportion | 58.4 | 17.8 | 9.6 | 6.9 | 4.4 | 1.2 | 0.8 | 0.5 | 0.2 | 0.1 |
| Cumulative | 58.4 | 76.3 | 85.9 | 92.8 | 97.2 | 98.4 | 99.2 | 99.7 | 99.9 | 100 |

The distance between the treatment interactions (cultivar × location × year) and the biplot origin is an exclusive proportion. In this manner, treatment interactions such as TH6 × BARI × 2015, TH6 × URF × 2015, TH6 × BARI × 2014, TH6 × NARC × 2014, TH6 × NARC × 2015 and TS3 × URF × 2014 are those that were present far away from the origin of the biplot, and were considered most diverse treatment interactions. Such treatment interactions may or may not be greater, but they may be useful for some important agronomic attributes.

### 3.7. Micronutrients

Zinc, boron and manganese treatments considerably amplified ($p < 0.01$) the growth and yield-related traits of sesame (Table 2). However, the increase over the control was 12.2%, 17.4% and 8.4% in capsules plant$^{-1}$; 18.7%, 10.3% and 5.4% biomass yield; and 14%, 16.8% and 7.9% in seed yield for Zn, B and Mn applications, respectively. Application of zinc resulted in the maximum increase in vegetative growth traits such as PH, NPB and NSB plus biomass yield. Likewise, the maximum increase in all yield attributes including seed yield was detected for boron treatment followed by zinc treatment (without significant differences, except for capsules plant$^{-1}$).

## 4. Discussion

The sesame cultivars revealed differences in the expression of their morphological characteristics due to genetic variability and the prevailing environmental factors. Similar explanations were suggested by the previous findings [24,25]. Among the branched cultivars, SG-30 and TH-5 produced the highest seed yield because of their dominant genetic ability and yield stability in all growing conditions, mainly due to producing a higher number of capsules plant$^{-1}$. However, TH-6 is an unbranched and short stature cultivar which was found to produce the lowest number of capsules, biomass yield and seed yield, but other traits concerning the yield of this cultivar were dramatically higher. Although the yield potential of TH-6 was moderately higher than other cultivars, if grown in favorable environmental conditions, it was highly susceptible to excessive soil moisture and high relative humidity, and reasonably tolerant to pests and diseases [26].

The results are the pooled means of sesame cultivars across three locations with different environmental conditions, particularly the amount and pattern of rainfall received during the cropping period. In addition, the sesame genotypes are unstable and diverse in their trait expression with changes in the growing environment and seasons (years) [27]. Therefore, sesame responded differentially in two growing seasons in terms of vegetative growth traits. The results of the current research are in agreement with previous work [24]. In the present study, the first year received comparatively higher rainfall during the growing period across all locations (Figure 1). Therefore, on average, growth attributes including biomass yield were higher during first year, which might be due to optimum soil moisture availability, adequate nutrient uptake and appropriate assimilate translocation into plant parts, which might have increased the photosynthetic capacity and resulted in higher vegetative growth.

The distinct effects of sesame in different environmental conditions could be attributed to irregular rainfall, soil and the phenotypic expression of the cultivars. Optimum soil moisture with proper soil aeration guaranteed an adequate oxygen supply, which regulated root respiration and the absorbance of nutrients. Similarly, optimum soil moisture availability owing to moderate and evenly distributed rainfall during the cropping period at BARI and URF ensured the higher growth and yield of sesame. Besides, the additional supply of macronutrients sustained the nutrient flow and high photosynthetic capacity with maximum assimilate supply to the sink that contributed to development, which might have ensured maximum growth and yield. However, sesame is highly prone to drought in several physiological stages but is principally vulnerable to heavy rainfall and waterlogging [28], which could hinder crop growth and yield. It is usually grown in overly dry areas with precipitation of 300–600 mm [29]. Excessive soil moisture limits the soil aeration and

oxygen supply, which restricts root respiration and nutrient uptake [30]. At NARC, during later growth stages, high soil moisture due to heavy rainfall in the months of September 2014 and October 2015 (Figure 1) might have reduced the growth and development of sesame and resulted in drastic yield reductions at this site. Such stress conditions might cause restricted plant metabolism, poor plant–water–nutrients relationships, limited photosynthetic capacity, and a slower crop growth rate and dry matter accumulation. In addition, heavy precipitation can cause nutrient leaching, leading to low productivity of the sesame crop. A considerable decline in the growth and yield of sesame due to excess soil moisture was also reported by [31].

Zinc and boron were found to be deficient in the soils of the experimental sites, while manganese was at a marginal level (Table 1). Therefore, the application of deficient micronutrients along with an adequate supply of macronutrients enhanced sesame growth and yield. Moreover, the increase in growth characteristics might be due to the critical role of zinc in hormone metabolism, protein and enzyme synthesis, and enzymatic activities [32], also increasing plants' tolerance to biotic and abiotic stresses, thus improving plant growth [33] and ultimately higher biomass. Additionally, boron is essentially required for the growth and development of a plant's reproductive parts [15,34]. Furthermore, boron improves the chances of fruit setting during pollen tube growth and germination [35]. Consequently, the highest results for NC, CL CW and 1000 SW were observed for boron application. Similarly, manganese is fundamentally crucial for plant metabolic activities and development, and it aids in chlorophyll formation and increases the accessibility of calcium and phosphorus [36]. Therefore, manganese application might have increased the growth and yield of sesame. The results of current study are parallel to the findings of previous research [15,17,37,38]. On the contrary, restricted growth and yield in the control treatments was due to a deficiency in micronutrients. For example, internode shortening and short height occurred as a result of reduced growth hormone development due to Zn deficiency [15]. Sesame is also very sensitive to boron deficiency. Its absence during the flowering stage decreases pollen viability and causes stamen and pistil abortion, resulting in poor seed set and low seed yield due to malformed branching [34]. Moreover, owing to boron deficiency, activation of enzymatic and non-enzymatic oxidation increased quinine concentrations and polyphenol oxidase, which are perilous for plant growth and development [39].

## 5. Conclusions

It can be concluded that TS-5 produced the maximum yield at URF, and SG-30 produced the maximum yield at BARI and NARC. These cultivars are recommended for general cultivation in the Pothwar region. Micronutrient fertilizer application produced significant increases in the productivity of sesame. Application of micronutrient fertilizers (10 kg $ZnSO_4$ ha$^{-1}$, 10 kg borax ha$^{-1}$ and 5 kg ha$^{-1}$ $MnSO_4$) along with the recommended dose of macronutrients significantly improved the productivity of sesame. The reduced production of sesame under high rainfall areas showed the low suitability of sesame under such conditions. Hence, low- and medium-rainfall areas with medium texture and well-drained soils are best suited for sesame cultivation.

**Author Contributions:** A.M. and F.u.H. conceived of the idea. M.M.K. conducted the experiment and conducted the literature review. A.Q., M.S.A. and Z.H.S. provided technical expertise. H.A. and S.Y. helped in statistical analysis. G.C. proofread and provided intellectual guidance. All authors read the first draft, helped in revision and approved the article. All authors have read and agreed to the published version of the manuscript.

**Funding:** This work was carried out with the support of the "Cooperative Research Program for Agriculture Science and Technology Development (Project No. PJ01581201)" Rural Development Administration, Republic of Korea.

**Institutional Review Board Statement:** Not applicable.

**Informed Consent Statement:** Not applicable.

**Data Availability Statement:** The data presented in this study are available upon fair request from the corresponding author.

**Conflicts of Interest:** The authors declare no conflict of interest.

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
