# Peer review of "Allometric Expression of Sesame Cultivars in Response to Various Environments and Nutrition"

_agriculture, doi:10.3390/agriculture11111095_

Round 1

Reviewer 1 Report

The manuscript was written very well. It is grammatically sound.

The experiment was conducted with correct design. However, the data analysis need to be improved. Please mention the model you used to conduct the anova, mention if the data were normally distributed, you may show histogram.

Lot of figures were put in the manuscript to cover all the data analysis. This type of strategy is good for writing thesis. However, if you are putting a manuscript, you need to be more summarized and be to the point. You need to structure a story of your experiment. You listed all data analysis in a separate header. Result and discussion part is just results, I would recommend having discussion part separately where you can discuss the story and connect the dots using your knowledge. I would recommend revising it and present in a way it attract the audience rather than list the results.

Correlation analysis is missing in the manuscript.

Line 50: period annotation missing at the end of sentence

Author Response

Response to Reviewer # 1

Dear reviewer, we are grateful to you for your comments and suggestions for the improvement of our research manuscript. We have tried our best to revise the manuscript in light of your comments.

Comment 1:   The manuscript was written very well. It is grammatically sound.

Response:       We are grateful for your compliment and encouragement.

Comment 2:   The experiment was conducted with correct design. However, the data analysis need to be improved. Please mention the model you used to conduct the anova, mention if the data were normally distributed, you may show histogram.

Response:       The ‘Statistical analysis’ is more elaborated for better understanding regarding the data analysis, also the model is mentioned which is used for ANOVA (see Page 4, Paragraph 2, Line # 136-146). Highlighted with yellow color.

Comment 3:   Lot of figures were put in the manuscript to cover all the data analysis. This type of strategy is good for writing thesis. However, if you are putting a manuscript, you need to be more summarized and be to the point. You need to structure a story of your experiment. You listed all data analysis in a separate header. Result and discussion part is just results, I would recommend having discussion part separately where you can discuss the story and connect the dots using your knowledge. I would recommend revising it and present in a way it attract the audience rather than list the results.

Response:       Yours suggestion and recommendation were taken seriously and found very helpful for improvement and presentation of manuscript. Therefore, we revised the entire manuscript in light of your proposition. So, following changes and improvements were carried out.

  • Results were summarized by providing necessary data information.
  • Data analysis were removed as separate header, moreover, results are following with necessary information, also sufficient data is the part of table 2.
  • Results and discussion parts are separated and kept accordingly. (see from Page 4, Line # 147 and from Page 13, Line # 306). Highlighted with yellow color.

Comment 4:   Correlation analysis is missing in the manuscript.

Response:       Correlation analysis among the agronomic traits was included in manuscript. (See Page 4, Line # 142-144 and Page 11, Line # 249). Highlighted with yellow color.

Comment 5:   Line 50: period annotation missing at the end of sentence.

Response:       The missing period annotation is added (see Page 2, Line # 50). Highlighted with yellow color.

Reviewer 2 Report

The paper is quite interesting for the allometric expression of Sesame cultivars (four) to three different environments and supplemented micronutrients (Zn, B, Mn) nutrition. However, I would like to make some suggestions that are listed below.

Title: It would be advisable to refer Sesame Cultivars instead Sesame Genotypes, as they are really seedlings cultivars plants and not 4 different genotypes, each one produced by vegetative propagation from 4 different mother-plants.

Abstract: The abstract is quite clear. However, it is not referred before results information, that the experiments were performed during two followed years. This information should be given before results.

Keywords: The keywords should be listed in a alphabetic order, moreover should not contain words that are already in the Title (environment; sesame)

Materials and Methods: At page 2, line 101, the authors should refer the meaning of DAP;

Results

At page 5, lines 179-181, the authors refer “Likewise, maximum increase in all yield attributes including seed yield was observed for boron application followed by zinc.” It should be advisable to add the following information “by zinc (without significant differences, except for the capsules plant-1)”.

Years - At page 6, lines 218-220, the authors refer “Years: Year means showed differential response for observed parameters (Table 2). Plant height (111.4 cm), number of primary branches (4.76), secondary branches (3.77) and biomass yield (5657 kg ha-1) were significantly higher during first year (2014).    ”. However, it should be advisable that authors refer that no significant differences were found for seed yield (kg/ha) due to year, as main factor. Moreover, comparing with other main factors of this study, year was the unique not showing significant differences for seed yield and with the lowest SE (9.2 kg/ha). Therefore, the author’s statement at page 6, lines 227-228 “Therefore, differential responses of growth and yield traits of sesame were observed during two growth seasons” is not completely true. This statement should be corrected. Furthermore, the cultivars showed to be the most relevant main factor for seed yield, followed by locations factor, without significant differences due to year effect and with a clear positive effect with the micronutrient supplementary application.

Interactions - At page 6, lines 235-236, the authors refer “Graphical presentations of interactions for quantitative characters of sesame are shown in figures no. 2 - 10.” It should be advisable to refer “Graphical presentations of interactions (showing significant differences or just P≤ 0.01) for quantitative characters of sesame are shown in figures no. 2 - 10.”

Some suggestions for Figures 2 – 10 – pages 7 -11

  • Figure captions, for example: Figure 2. Plant height (cm) of sesame at three locations and according the four cultivars tested (pooled over micronutrients) during 2014 and 2015.
  • Titles for y axis.
  • It is advisable add letters over columns to show significant differences among the conditions tested.

It should be advisable to add a Principal Component Analysis (PCA) to understand the interrelations among the main factors and the parameters observed

Author Response

Response to Reviewer # 2

Dear reviewer, we are grateful to you for your comments and suggestions for the improvement of our research manuscript. We have tried our best to revise the manuscript in light of your comments.

Comment 1:   The paper is quite interesting for the allometric expression of Sesame cultivars (four) to three different environments and supplemented micronutrients (Zn, B, Mn) nutrition.

Response:       We are grateful for your compliment and encouragement.

Comment 2:   Title: It would be advisable to refer Sesame Cultivars instead Sesame Genotypes, as they are really seedlings cultivars plants and not 4 different genotypes, each one produced by vegetative propagation from 4 different mother-plants.

Response:       The word ‘genotype’ is replaced with ‘cultivar’ in title of manuscript. (see Page 1, Line # 2). Highlighted with green color

Comment 3:   Abstract: The abstract is quite clear. However, it is not referred before results information, that the experiments were performed during two followed years. This information should be given before results.

Response:       The suggestion information is added in abstract “in two succeeding years (2014, 2015)”. (see from Page 1, Line # 18). Highlighted with green color.

Comment 4:   Keywords: The keywords should be listed in a alphabetic order, moreover should not contain words that are already in the Title (environment; sesame)

Response:       key words were revised with replacement and kept in order alphabetically. (See Page 1, Line # 33). Highlighted with green color.

Comment 5:   Materials and Methods: At page 2, line 101, the authors should refer the meaning of DAP;

Response:       Full form of DAP (di-ammonium phosphate) is added. (see Page 2, Line # 98). Highlighted with green color.

Comment 6:   Results: At page 5, lines 179-181, the authors refer “Likewise, maximum increase in all yield attributes including seed yield was observed for boron application followed by zinc.” It should be advisable to add the following information “by zinc (without significant differences, except for the capsules plant-1)”.

Response:       The explanation as suggested are added accordingly. (see Page 12, Line # 300-301). Highlighted with green color.

Comment 7:   Years - At page 6, lines 218-220, the authors refer “Years: Year means showed differential response for observed parameters (Table 2). Plant height (111.4 cm), number of primary branches (4.76), secondary branches (3.77) and biomass yield (5657 kg ha-1) were significantly higher during first year (2014)”. However, it should be advisable that authors refer that no significant differences were found for seed yield (kg/ha) due to year, as main factor. Moreover, comparing with other main factors of this study, year was the unique not showing significant differences for seed yield and with the lowest SE (9.2 kg/ha). Therefore, the author’s statement at page 6, lines 227-228 “Therefore, differential responses of growth and yield traits of sesame were observed during two growth seasons” is not completely true. This statement should be corrected. Furthermore, the cultivars showed to be the most relevant main factor for seed yield, followed by locations factor, without significant differences due to year effect and with a clear positive effect with the micronutrient supplementary application.

Response:       The upraised points were considered and the necessary rephrasing were carried out (see Page 5, Line # 170-171; Page 13, Line # 321-322). Highlighted with green color.

Comment 8:   Interactions - At page 6, lines 235-236, the authors refer “Graphical presentations of interactions for quantitative characters of sesame are shown in figures no. 2 - 10.” It should be advisable to refer “Graphical presentations of interactions (showing significant differences or just P≤ 0.01) for quantitative characters of sesame are shown in figures no. 2 - 10.”

Response:       The clarification is added as suggested. (see Page 5, Line # 178-179). Highlighted with green color.

Comment 9:   a) Figure captions, for example: Figure 2. Plant height (cm) of sesame at three locations and according the four cultivars tested (pooled over micronutrients) during 2014 and 2015. (b) Titles for y axis. (c) It is advisable add letters over columns to show significant differences among the conditions tested.

Response:       The following actions were taken as suggested (See Page 6-10, figures no. 2-10, Line # 182-204). Highlighted with green color.

  • Figure captions were rephrased providing necessary clarification as suggested.
  • Titles for Y-axis were incorporated in all figures
  • Letters over column bars were included in all figures

Comment 10: It should be advisable to add a Principal Component Analysis (PCA) to understand the interrelations among the main factors and the parameters observed

Response:       Principal Component Analysis (PCA) along with principal component matrix and PCA bi-plot are included in manuscript. (See Page 4, Line # 144-146; Page 11-13, Line # 263-304). Highlighted with green color.

Reviewer 3 Report

The manuscript submitted for review meets the requirements for scientific contribution and content. A very important factor influencing the yield and phenotypic features of plants are the genetic conditions of cultivars that are sensitive to changing environmental conditions, as well as fertilization. Therefore, it is advisable to study the reaction of sesame cultivars to various environmental conditions and the influence of zinc, boron and manganese on the growth and yielding characteristics of sesame. The authors of the work carried out extensive experience related to it, and properly adopted the research methods. In some of the results and discussions, the results are correctly presented in the table and graphically, which is more readable with a lot of results. The influence of the genetic determinants of the cultivar, the micronutrients used, the location and years of research and their interactions were correctly explained, and the results of studies by other authors were referred to.

Several errors can be noted that do not affect the value of the work presented:

- line 80, 193- reference is missing from Singaravel et al., 2002;

- line 138- Statistical analysis- in the result part we also have the significance level 0.01 - missing in the methodology

- line 220- error of the result- should be 2657 (tab. 2) and not 5657 kg ha-1

- in table 2 and line 299, instead of biological yield, I would enter biomass yield as in the description below

Author Response

Response to Reviewer # 3

Dear reviewer, we are grateful to you for your comments and suggestions for the improvement of our research manuscript. We have tried our best to revise the manuscript in light of your comments.

Comment 1:   The manuscript submitted for review meets the requirements for scientific contribution and content. A very important factor influencing the yield and phenotypic features of plants are the genetic conditions of cultivars that are sensitive to changing environmental conditions, as well as fertilization. Therefore, it is advisable to study the reaction of sesame cultivars to various environmental conditions and the influence of zinc, boron and manganese on the growth and yielding characteristics of sesame. The authors of the work carried out extensive experience related to it, and properly adopted the research methods. In some of the results and discussions, the results are correctly presented in the table and graphically, which is more readable with a lot of results. The influence of the genetic determinants of the cultivar, the micronutrients used, the location and years of research and their interactions were correctly explained, and the results of studies by other authors were referred to.

Response:       We are so grateful for your in-depth appreciation, compliment and it is very encouraging.

Comment 2:   line 80, 193- reference is missing from Singaravel et al., 2002;

Response:       The missing reference is added in revised manuscript. (see Page 15, Line # 425-426). Highlighted with sky-blue color.

Comment 3:   line 138- Statistical analysis- in the result part we also have the significance level 0.01 - missing in the methodology.

Response:       Initially, when liner model was applied through computer software, p-values were obtained from ANOVA table, which illustrated the significance level of main factors and interacted-treatments. Whereas, several values showed highly significant (p<0.01) and these were considered for data results. In the next step, LSD test was applied with 5% probability (alpha value 0.05), which showed significance of differences among the means, and illustrated as letters in tables and figures.

Comment 4:   line 220- error of the result- should be 2657 (tab. 2) and not 5657 kg ha-1

Response:       The mentioned error is removed text in manuscript. (see from Page 5, Line # 172). Highlighted with sky-blue color.

Comment 5:   in table 2 and line 299, instead of biological yield, I would enter biomass yield as in the description below.

Response:       Action is taken as suggested. ‘biological yield’ is replaced with ‘biomass yield’ in revised manuscript. (Highlighted with sky-blue color).

Round 2

Reviewer 1 Report

Thanks addressing all my edits. Good luck

Reviewer 2 Report

The corrections and suggestions were performed as requested